# Neutron crystallography reveals mechanisms used by *Pseudomonas aeruginosa* for host-cell binding

Lukas Gajdos [1,2,3], Matthew P. Blakeley [4], Michael Haertlein[1,2], V. Trevor Forsyth[1,2,5,6,7], Juliette M. Devos [1,2✉] & Anne Imberty [3✉]

The opportunistic pathogen *Pseudomonas aeruginosa*, a major cause of nosocomial infections, uses carbohydrate-binding proteins (lectins) as part of its binding to host cells. The fucose-binding lectin, LecB, displays a unique carbohydrate-binding site that incorporates two closely located calcium ions bridging between the ligand and protein, providing specificity and unusually high affinity. Here, we investigate the mechanisms involved in binding based on neutron crystallography studies of a fully deuterated LecB/fucose/calcium complex. The neutron structure, which includes the positions of all the hydrogen atoms, reveals that the high affinity of binding may be related to the occurrence of a low-barrier hydrogen bond induced by the proximity of the two calcium ions, the presence of coordination rings between the sugar, calcium and LecB, and the dynamic behaviour of bridging water molecules at room temperature. These key structural details may assist in the design of anti-adhesive compounds to combat multi-resistance bacterial infections.

[1] Life Sciences Group, Institut Laue-Langevin, 71 Avenue des Martyrs, 38000 Grenoble, France. [2] Partnership for Structural Biology (PSB), 71 Avenue des Martyrs, 38000 Grenoble, France. [3] Université Grenoble Alpes, CNRS, CERMAV, 38000 Grenoble, France. [4] Large Scale Structures Group, Institut Laue-Langevin, 71 Avenue des Martyrs, 38000 Grenoble, France. [5] Faculty of Natural Sciences, Keele University, Staffordshire ST5 5BG, UK. [6] Faculty of Medicine, Lund University, BMC Biomedical Centre, 221 00 Lund, Sweden. [7] LINXS Institute for Advanced Neutron and X-ray Science, IDEON Building: Delta 5, Scheelvagen 19, 223 70 Lund, Sweden. ✉email: devosj@ill.fr; anne.imberty@cermav.cnrs.fr

The bacterium *Pseudomonas aeruginosa* is an opportunistic pathogen responsible for lung infections in cystic fibrosis patients and in immunocompromised patients, especially in hospital environments. It is particularly problematic in intensive care units. It is armed with an arsenal of virulence factors and antibiotic resistance determinants[1] and has been identified as the number one priority, in the list of antibiotic-resistant bacteria by the WHO in 2017, in order to prioritize the efforts for the development of new antibiotics[2]. *P. aeruginosa* produces two soluble lectins[3,4] LecA (PA-IL) and LecB (PA-IIL) specific for galactose and fucose, respectively, that are involved in the bacterial pathogenicity, adhesion and biofilm formation[5–7]. LecB is of special interest, since it is present only in a few opportunistic bacteria related to *Pseudomonas*[8]. LecB is a quorum-sensing virulence factor[9]. In addition to its role in biofilm stabilization[6], it is directly involved in lung colonization in a mouse infection model[7,10] and in inhibition of wound healing[11,12]. The structure of LecB, and a few related LecB-like proteins, have a unique binding site with two closely located calcium ions (3.76 Å apart) that are directly involved in the sugar binding through the coordination of three hydroxyl groups of the carbohydrate ligand[13] (Fig. 1). The two calcium ions contribute to the receptor specificity since they coordinate monosaccharides with the stereochemistry of two equatorial and one axial hydroxyl group present in fucose and mannose. These two calcium ions are also believed to play a role in the unusually strong affinity of LecB for fucose, which is in the low micromolar range, as compared to the millimolar affinity usually observed for lectin/monosaccharide interactions[14]. Carbohydrate chemists have proposed several fucose or mannose derivatives as high-affinity ligands[15] and some of these are able to inhibit biofilm growth, either as small molecules[16] or as multivalent compounds[17].

A high-resolution crystal structure of LecB has been obtained previously[14], but the role of the two calcium ions in the high-affinity binding of fucose remains unclear and further structural analyses are required. Neutron macromolecular crystallography is the technique of choice for determining the positions of hydrogen atoms and the protonation states of active groups[18–21] that are involved in the catalytic activity of enzymes or the binding of a ligand to a macromolecule[22–25]. Moreover, neutrons do not cause any radiation damage to the biological sample so the structure determination and data collection can be performed at room temperature (RT), giving a more accurate view of local flexibility and water mobility at biologically relevant temperatures[26,27].

In the present work, perdeuterated fucose, produced through the use of engineered bacteria, has allowed, for the visualization of all hydrogen atoms (deuterium isotope) in the complex between a lectin from a human pathogen and its targeted sugar. We demonstrate the presence of several mechanisms underlying the observed high-affinity binding, including a low-barrier hydrogen bond, cooperative rings of hydrogen bonds and coordination contacts resulting in unique charge delocalization, and the mobility of trapped water molecules. Such structural details are of crucial importance for a more precise understanding of the protein/sugar/calcium triplet, and indeed, for the design of glycomimetic drugs that would reach higher affinity.

## Results

**Solving the neutron structure of perdeuterated LecB/fucose complex.** In order to determine the position of hydrogen atoms in the LecB/fucose complex, both molecules were first obtained in perdeuterated form. Hence, in the neutron analysis, hydrogen atoms are seen as deuterium, but for clarity in the following sections, are hereafter referred to as "hydrogen atoms". Recombinant perdeuterated LecB (D-LecB) was produced in an *Escherichia coli* high cell-density culture in the presence of $D_2O$ and glycerol-$d_8$ after adaptation of cells[28]. Perdeuterated fucose (Fuc-$d_{12}$) was obtained using recently described synthetic biology methods for which *E. coli* strains re-engineered for their carbohydrate metabolism were adapted for growth in deuterated media[29]. Co-crystals with volume of ~0.1 $mm^3$ were obtained over a period of ~4 weeks. The neutron structure of D-LecB complexed with Fuc-$d_{12}$ was obtained at 1.90 Å resolution, and was jointly refined against both neutron and X-ray data (1.85 Å) collected from the same crystal at RT. The data collection and refinement statistics are shown in Table 1. As described previously, the overall fold of the LecB monomer is a β-sandwich composed of two curved sheets consisting of five and four antiparallel β-strands, respectively[13] (Supplementary Fig. 1). Since the general features of the structure of LecB and the binding site with fucose have been well documented, the present work focuses solely on the position of hydrogen atoms and the effect of RT analysis.

**Hydrogen atoms at monomers interface.** The excellent quality of the neutron scattering length density maps (hereafter referred to as "neutron maps") allows clear observation of the aliphatic amino acids and hydrogen bonds between adjacent β-sheets in the monomer. The tetramerization that occurs through the antiparallel association of β-strands from each dimer with their counterparts in the other dimer, involves mostly hydrogen bonds that could be directly visualized from the neutron maps. Furthermore, the use of perdeuterated protein has allowed the

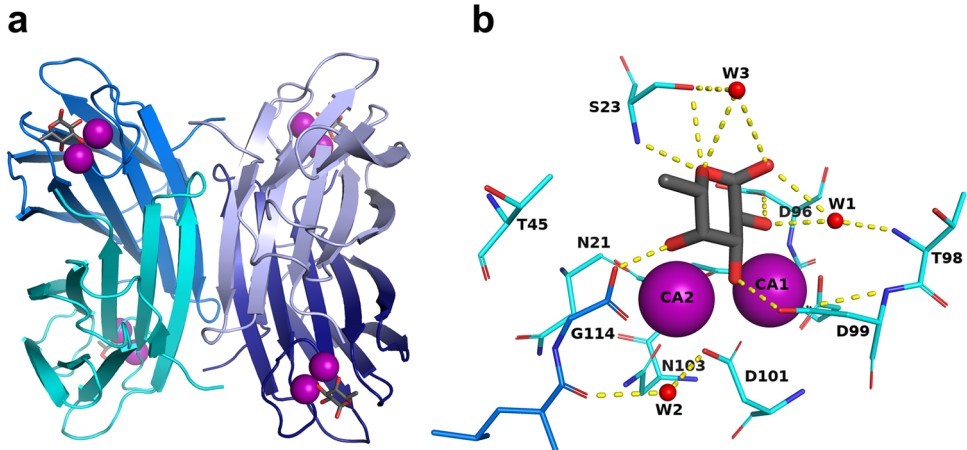

**Fig. 1 Structure of LecB lectin in complex with fucose. a** Tetramer of LecB (PDB code: 1GZT). **b** Fucose-binding site. Calcium ions and water molecules are represented by purple and red spheres respectively. Hydrogen bonds are shown as yellow dashed lines.

**Table 1 Room temperature neutron data collection, 100 K X-ray data collection and refinement statistics for the perdeuterated LecB/fucose complex.**

| | D-LecB Fucose-d$_{12}$ | D-LecB Fucose-d$_{12}$ |
|---|---|---|
| Data collection | | |
| Temperature | RT | 100 K |
| Neutrons | | |
| Instrument | LADI-III, ILL | |
| Wavelengths (Å) | 2.8–3.8 | |
| Detector | Image plate | |
| Resolution (Å) | 43–1.90 (2.08–1.90) | |
| Spacegroup | $P2_1$ | |
| Unit cell parameters | | |
| $a, b, c$ (Å) | 52.9, 73.9, 55.0 | |
| $\alpha, \beta, \gamma$ (°) | 90, 94.6, 90 | |
| $R_{merge}$ ($I$) (%) | 21.1 (38.5) | |
| $R_{pim}$ ($I$) (%) | 10.9 (38.5) | |
| Mean $I/\sigma$ ($I$) | 5.5 (2.1) | |
| Completeness (%) | 73.8 (61.7) | |
| Multiplicity | 3.7 (3.9) | |
| No. of unique reflections | 24,478 (4860) | |
| Crystal size (mm$^3$) | 0.1 | |
| X-rays | | |
| X-ray source | GeniX 3D Cu High Flux (Xenocs), IBS | Proxima-1, SOLEIL |
| Wavelength (Å) | 1.5418 | 0.97856 |
| Detector | Mar 345 (marXperts) | EIGER-X 16 M (Dectris Ltd.) |
| Resolution (Å) | 33–1.85 (1.89–1.85) | 40–0.90 (0.92–0.90) |
| Unit cell parameters | | |
| $a, b, c$ (Å) | 52.9, 73.9, 55.0 | 52.6, 73.0, 55.2 |
| $\alpha, \beta, \gamma$ (°) | 90, 94.6, 90 | 90, 94.6, 90 |
| $R_{merge}$ ($I$) (%) | 9.7 (29.8) | 4.3 (141.6) |
| $R_{meas}$ ($I$) (%) | 10.4 (36.2) | 4.7 (171.6) |
| CC1/2 (%) | 99.5 (84.7) | 100 (63.2) |
| Mean $I/\sigma$ ($I$) | 11.3 (4.2) | 16.3 (0.6) |
| Completeness (%) | 96.6 (70.4) | 87.6 (25.6) |
| Multiplicity | 7.0 (2.5) | 6.1 (2.4) |
| No. of unique reflections | 34,855 (1586) | 270,061 (5821) |
| Refinement | | |
| Resolution range X-ray (Å) | 27.41–1.85 | 36.53–0.90 |
| Resolution range neutron (Å) | 42.92–1.90 | |
| Reflections (used), X-ray | 34,734 | 526,848 |
| Reflections (test), X-ray | 1736 | 25,913 |
| Reflections (used), neutron | 24,283 | |
| Reflections (test), neutron | 1195 | |
| $R_{work}$ (%), X-ray | 10.4 | 11.6 |
| $R_{free}$ (%), X-ray | 14.2 | 13.1 |
| $R_{work}$ (%), neutron | 19.1 | |
| $R_{free}$ (%), neutron | 24.6 | |
| No. of atoms (protein) | 3558 | 3573 |
| No. of water molecules | 364 | 774 |
| RMSD in bond lengths (Å) | 0.010 | 0.008 |
| RMSD in bond angles (°) | 1.269 | 1.107 |

**Table 1 (continued)**

| | D-LecB Fucose-d$_{12}$ | D-LecB Fucose-d$_{12}$ |
|---|---|---|
| Average B factors (Å$^2$) | | |
| Protein | 20.3 | 11.0 |
| Ligand | 18.3 | 12.6 |
| Ramachandran statistics | | |
| Favored (%) | 97.3 | 96.2 |
| Allowed (%) | 2.7 | 3.8 |
| Outliers (%) | 0.0 | 0.0 |
| All-atom clashscore | 1.4 | 5.0 |
| PDB code | 7PRG | 7PSY |

observation of the contacts involved in tetramer formation, such as the hydrogen bond between N-ter Ala1 of chain A and Asp75 of chain C, and the hydrophobic interactions between the methyl groups of the aliphatic amino acid side chains of Ala1 (chain A), Val77 (chain C) and Thr84 (chain D) (Supplementary Fig. 1b). The orientation of key water molecules and polar amino acid side chains could also be unambiguously determined from the neutron analysis (Supplementary Fig. 1c).

**Visualization of hydrogen atoms in perdeuterated fucose.** In all four chains, the fucose ring adopts the stable $^1C_4$ chair conformation, with the α-configuration at the anomeric position. The use of perdeuterated fucose provided high-quality neutron density maps showing positive peaks for the deuterium atoms on the ring carbon atoms, the fucose methyl group, and the four hydroxyl groups (Fig. 2). In all four monomers, the neutron map clearly shows the hydrophobic contacts between the methyl group on the C6 position of fucose and the -CH$_2$ and methyl groups of Ser23 and Thr45, respectively (Fig. 2a). The deuterium atoms in the -CH$_3$ group create a pyramid-shaped neutron density and are in the most stable staggered conformation with respect to the aliphatic deuterium atom on C5.

The deuterium atoms attached to the four hydroxyl groups can be clearly visualized pointing away from the metal ions, establishing hydrogen bonds with the amino acid residues in the binding site. Peaks in the neutron density map can be observed for all of the hydrogen bonds established between the fucose and the protein (Fig. 2 and Supplementary Table 1). The Fuc-OD2 hydroxyl group donates its deuterium atom to OD1 of Asp96, Fuc-OD3 establishes a hydrogen bond with the side chain OD2 oxygen of Asp99, and Fuc-OD4 donates a deuterium atom to the OXT oxygen of the carboxyl group of Gly114 of the neighboring monomer. The ring oxygen of fucose accepts a hydrogen bond from the amide backbone of Ser23.

For comparison, the 0.9 Å resolution X-ray structure, obtained at 100 K, was also refined, showing the location of several hydroxyl groups in the m$F_o$-DF$_c$ omit map electron density (Supplementary Fig. 2). However, some hydrogen atoms could not be located and the overall quantity of information is lower. Nevertheless, the analyses of the RT (neutron/X-ray) and 100 K (X-ray) structures are also of interest for a comparison of thermal motion. In Supplementary Fig. 3, it can be seen that two of the four monomers in the tetramer are more stable, and that there is less thermal motion than for the others. This observation may have implications for the hydrogen bonds within the active site and will be discussed later.

**Coordination of calcium ions.** LecB and related lectins have unique carbohydrate receptors with two adjacent calcium ions in the binding site directly involved in sugar binding. Calcium atoms have a neutron scattering length of 4.70 fm, which is lower than

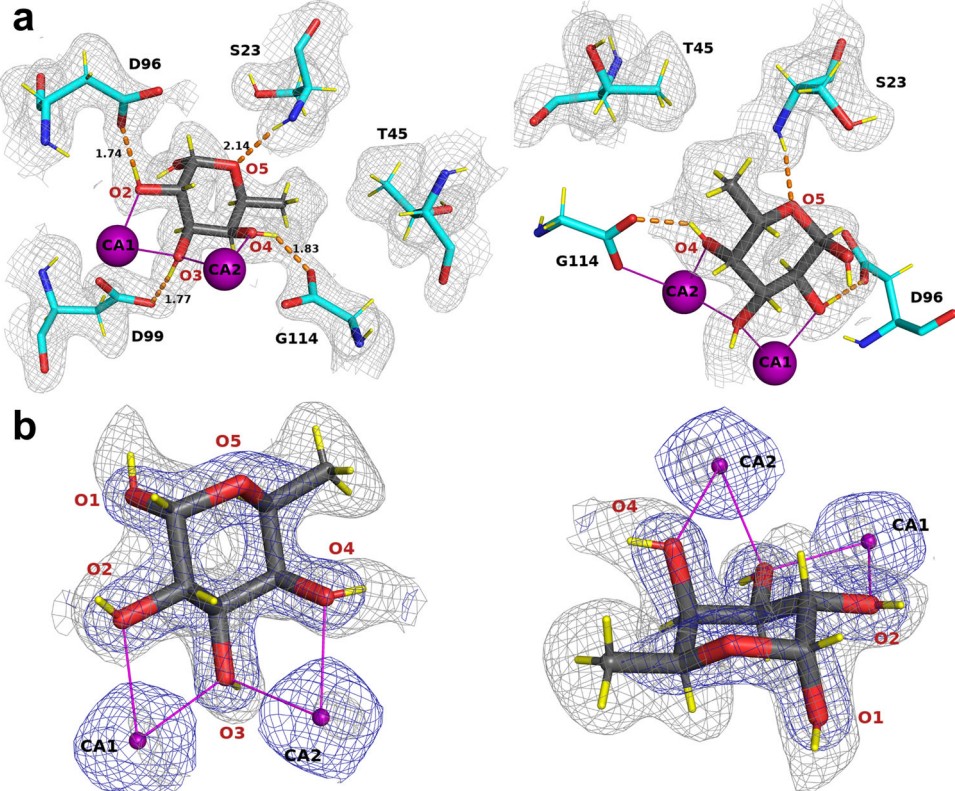

**Fig. 2 Fucose-binding site in the neutron structure of the D-LecB/Fuc-d$_{12}$ complex. a** Two different orientations of the same fucose-binding site (chain B). Fucose is represented as dark gray sticks and protein as cyan sticks. The calcium ions are shown as purple spheres and deuterium atoms are shown as yellow sticks. The hydrogen bonds and metal coordination are shown as orange dashed lines and as solid purple lines, respectively. The distances are given in Å. The 2m$F_o$-D$F_c$ neutron map (gray mesh) is contoured at 0.8$\sigma$. Amino acids involved in coordination of the calcium ions were omitted for clarity (Asn21, Asn103, Asp101, Asp104, Glu95). The first view (left) highlights the hydrogen-bonding network between fucose and the protein. The second view (right) shows the hydrophobic contact between methyl group of fucose and methyl and -CH$_2$ groups of the Thr45 and Ser23 side chains, respectively. **b** The 2m$F_o$-D$F_c$ neutron map (gray mesh) and 2m$F_o$-D$F_c$ X-ray map (blue mesh), contoured at 1.4$\sigma$ and 2.2$\sigma$, respectively.

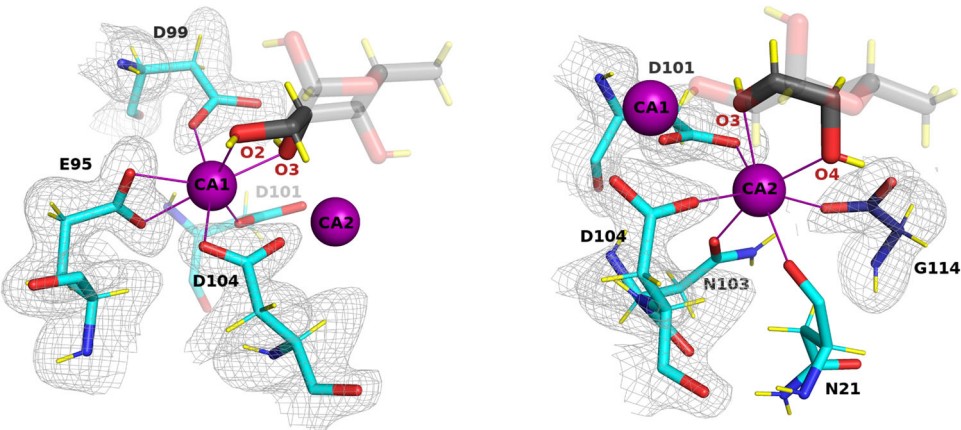

**Fig. 3 Calcium ions environment and protonation state of acidic residues.** The 2m$F_o$-D$F_c$ neutron scattering length density (gray mesh) is contoured at 0.8$\sigma$. Calcium ions are shown as purple spheres and metal coordination as solid purple lines (binding site in chain B is shown).

that for C, O, N and D; hence their precise locations were refined from the X-ray data, confirming the close distance of 3.76 Å ± 0.02[14]. Each calcium is hepta-coordinated (Fig. 3), with the involvement of three oxygen atoms of the fucose ligand (2 of these for each calcium ion), oxygen atoms from two asparagine residues (side chain of Asn103 and main chain of Asn21) and the carboxylate atoms of five amino acids (aspartates 99, 101, and 104, glutamate 95 and the C-terminal group of Gly114 from the

neighboring monomer). The protonation states of these groups and of Asp96 that binds to fucose is a crucial question for an understanding of the mechanism of high-affinity binding. This can be best analyzed from the neutron density map of the fully deuterated complex that was used in this study. No hydrogen atoms could be detected on the acidic groups of the binding site, indicating that these are all negatively charged (Fig. 3). The resulting total excess of six electrons is not fully compensated by

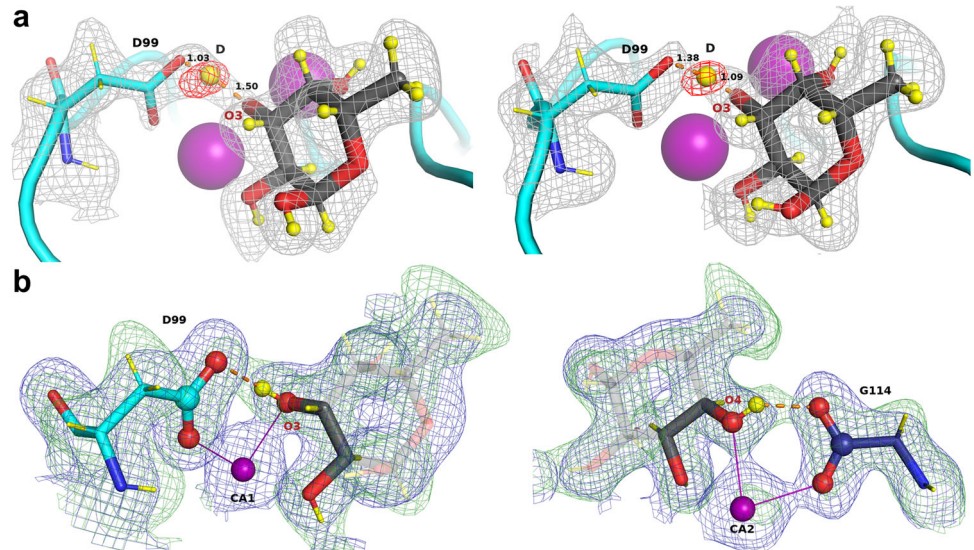

**Fig. 4 Mechanistic details of the proposed synergistic interactions involving hydrogen bonds and metal coordination. a** A low-barrier hydrogen bond in chain A (left) and D (right) located from the omit difference $mF_o$-D$F_c$ neutron map (red mesh) in the vicinity of O3 of fucose. **b** Two six-membered rings (illustrated by atoms in sphere representation) formed by proposed synergistic interactions between fucose, calcium, and LecB. Hydrogen bonds are shown as orange dashed lines and metal coordination as solid purple lines. Deuterium atoms are shown as yellow spheres and yellow sticks. The $2mF_o$-D$F_c$ neutron map (green mesh) is contoured at $1\sigma$. The $2mF_o$-D$F_c$ X-ray map (blue mesh) is contoured at $1.2\sigma$.

the charge on the calcium ions and therefore the binding pocket of LecB has a strong net negative charge.

**Effect of deuteration on structures and thermodynamics.** The structures of the perdeuterated LecB/fucose complex can be directly compared to the previously reported hydrogenated LecB/Fuc complexes[14] since the crystals were obtained using similar crystallization conditions. The structures are very similar with an RMSD of 0.13 Å over the whole tetramer main chain. The thermodynamics of fucose binding to LecB was also compared between the fully hydrogenated and fully deuterated systems, in $H_2O$ and $D_2O$, respectively (Supplementary Fig. 4). No significant differences could be observed between the two studies with an enthalpy of binding $\Delta H$ of $-30.45$ ($\pm 0.05$) kJ/mol for the hydrogenated system and $-30.95$ ($\pm 0.15$) kJ/mol for the deuterated one. The dissociation constants were also almost identical with $K_D$ values of 7.64 ($\pm 0.15$) μM and 7.96 ($\pm 0.42$) μM, respectively. This confirms that, for this particular protein/carbohydrate complex, the deuterated system is closely isomorphous with the unlabeled one, and that the results obtained for the visualization and the location of the deuterium atoms can be safely extrapolated to hydrogen atoms.

**Role of calcium ions in promoting a low-barrier hydrogen bond.** The analysis of the neutron structure shown above has brought insights to the mechanistic basis underlying the unusually high affinity of LecB toward fucose—providing information that could not have been obtained from X-ray diffraction data alone.

Among the four hydrogen bonds between fucose and LecB, the three involving the carboxylate groups of amino acids coordinating fucose can be described as short (donor-acceptor distance <2.6 Å) while the fourth one from the Ser23 main chain amine has a more standard length (distance >2.9 Å)[30]. The nature of the short H-bonds was further investigated in order to determine the location of the deuterium atoms while removing the geometric constraints imposed during structure refinement. The $mF_o$-D$F_c$ neutron density maps, for which the deuterium atoms of the fucose hydroxyl groups were omitted, were therefore used to

locate the peaks of deuterium atoms for chains A and D that display lower B factors (Supplementary Fig. 3) and better quality maps (Supplementary Table 2 and Fig. 4a).

The positive density peaks in which deuterium atoms can be placed are close to oxygen atoms O2 and O4, and would correspond to short O-D distances of $0.85 \pm 0.09$ Å and $0.71 \pm 0.01$ Å, respectively, confirming a covalent link between deuterium and fucose oxygen atoms. The resulting hydrogen bonds, involving Fuc-O2/Asp96 and FucO4/Gly114, can therefore be described as very short but classical. In contrast, the hydrogen bond between Fuc-O3D and Asp99 shows an unusual character with a deuterium atom located almost equidistantly between the carboxylic side chain of Asp99 and the O3 atom of the fucose molecule (Fig. 4a and Supplementary Table 2) at $1.30 \pm 0.21$ Å away from O3 of fucose and $1.25 \pm 0.17$ Å OD2 of Asp99 with a very open O..D..O angle of $162.2 \pm 1.8°$. The very short distance between the donor and acceptor ($2.47 \pm 0.04$ Å) and, importantly, the delocalization of the hydrogen atom equally shared by two heteroatoms defines a low-barrier hydrogen bond[31–33]. Neutron maps are the most appropriate tool to visualize such bonds that have been observed not only in enzyme transition states, but also in protein-ligand complexes, such as in periplasmic phosphate binding protein[34]. The hydroxyl O3 of fucose is located at a very special location between the two calcium ions, with the deuterium pointing out toward Asp99; it is clear that the proximity of the two cations participates in the bond formation. It has been proposed that such bonds may be approximately six times stronger than classical ones[35].

**Occurrence of rings of contacts.** Three fucose oxygen atoms coordinate two calcium ions, with O2 coordinating the first calcium, O3 both calcium ions, and O4 coordinating the second calcium (Fig. 3). The fucose hydroxyl groups O2 and O4 form hydrogen bonds with the acidic groups of Asp96 and the C-terminus of Gly114 of the neighboring chain, respectively. Both interactions can be defined as very short hydrogen bonds (dist = $2.54 \pm 0.04/2.49 \pm 0.01$ Å)[30] although with no evidence of the delocalization of the deuterium atom. They participate strongly in generating the high binding affinity. Furthermore, the

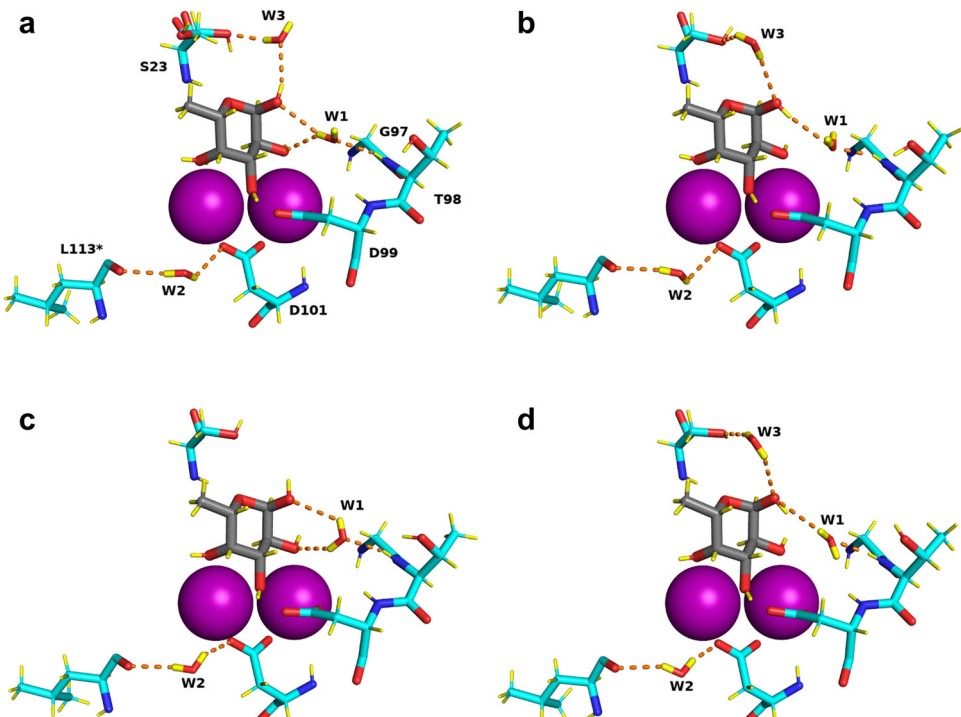

**Fig. 5 Water network in the fucose-binding sites of the perdeuterated LecB/Fuc-d$_{12}$ complex determined from the neutron structure. a–d** represent binding sites in chains A, B, C, and D, respectively. Calcium ions are represented by purple spheres. Hydrogen bonds are shown as orange dashed lines. Water molecules in the binding sites are shown in stick representation.

coordination of calcium and hydrogen bonds creates a network of contacts between two aspartate residues, each coordinating one calcium ion and accepting a hydrogen bond from a fucose hydroxyl group, forming two 6-membered rings (Fig. 4b). It is proposed that such rings have synergistic effect since previous quantum mechanical studies have suggested that, in presence of the fucose ligand, the charges of calcium delocalize very efficiently[14].

**Mobility of water network in ligand binding site.** Several conserved water molecules are commonly observed in LecB/Fuc complexes (PDB: 1GZT, 1OXC, 1UZV) and have established the existence of hydrogen-bonding networks between the protein, the ligand and the other ordered waters[13,14,36]. In the neutron structure, the water networks in each of the four monomers display small differences (Fig. 5 and Supplementary Fig. 5). The most conserved water molecule, W1, accepts a hydrogen bond from the amide backbone of Thr98 (2.07 Å ± 0.05 Å) in all monomers, but adopts different orientations and therefore donates hydrogen atoms to either oxygen O1 and O2 of fucose, or to both of them, depending on the chains. This water molecule has been considered as very stable on the basis of previous high-resolution X-ray studies[14]; however, the present neutron structure at RT demonstrates higher mobility.

This behavior is not general. When examining another conserved water site, i.e. W2 bridging between Asp101 side chain and Leu113 main carbonyl group, this water is very stable and is implicated in similar hydrogen bonds for the four monomers. The third water molecule W3 bridges the anomeric oxygen atom to the side chain of Ser23 and is not conserved in all LecB crystal structures. In the neutron structure, W3 is present in three out of the four monomers. The neutron density is less clear for W3 but it could still be oriented based on the 2m$F_o$-D$F_c$ neutron map, as could the orientation of both OD1 fucose hydroxyl group and the side chain of Ser23.

The neutron structure therefore provides a different view from that provided by the previous high-resolution structures - allowing the discrimination of the fine details of the water networks in the four subunits of LecB, as well as showing disorder in the water molecules at the interface between carbohydrate ligand and protein. The binding of fucose results in the release of water molecules strongly coordinated to the two calcium ions, and the protein/sugar interface does not create sites with strongly bound waters. While most protein-carbohydrate interactions display limited affinity because of unfavorable entropy barriers[37], in this particular case, the overall entropy contribution is predicted to be favorable, in agreement with experimental measurements by ITC on this family of two calcium containing lectins[38].

## Discussion

We report here the experimental determination of the directionality of the fucose hydroxyl groups and the protonation state of acidic residues in the carbohydrate-binding site of LecB lectin from human pathogen *Pseudomonas aeruginosa*. Previously, a high-resolution 100 K X-ray crystal structure (PDB: 1UZV) to 1 Å resolution allowed only one of the hydrogen atoms of O2-hydroxyl group to be oriented, pointing toward oxygen OD1 of the Asp96 side chain[14]. The protonation states of the acidic residues have not previously been determined and required us to use neutrons. The neutron analysis has provided evidence that all of the acidic residues coordinating the calcium atoms are not protonated. This is in line with the prediction based on the ab initio calculations of bond distances in acidic residues having different protonation states[14].

The unusually high affinity of LecB for fucose and fucosylated oligosaccharides can be rationalized by a synergy between short hydrogen bonds, including a low-barrier one, and coordination bonds between the sugar, protein and calcium ions. Moreover, the binding of fucose causes displacement of three tightly-bound

water molecules that are present in the sugar-free structure[36] to the bulk solvent that is accompanied by a favorable entropy of binding. Both the high enthalpic contributions from strong hydrogen bonds, a favorable entropic term, together with charge delocalization caused by the involvement of two close calcium ions, play a role in the high affinity. From RT neutron diffraction data it can be observed that water molecules which participate in the sugar binding display high mobility.

Medicinal chemistry, computer-based drug design or drug development, and in the future artificial intelligence structure-based drug design all require accurate protein structures with accurate models, inclusive of hydrogen atoms. These will all benefit from high levels of structural detail of the binding sites in order to give the best possible results for drug development[39]. The information presented here will be important in rational structure-based drug design of potent inhibitors of *Pseudomonas aeruginosa*. It is believed that this work and these types of approaches will make important contributions to the development of glycomimetics that could help deal with the resurgence of multi-drug resistance bacteria.

## Methods

**Protein expression**. The LecB lectin was expressed in *Escherichia coli* BL21(DE3) bacteria harboring a pET29b(+)-pa2l plasmid with a kanamycin-resistance gene. All cultures were grown at 37 °C with shaking at 180 rpm and were supplemented with 50 µg ml⁻¹ kanamycin.

**Adaptation to $D_2O$ and deuterated glycerol-$d_8$**. Precultures of the LecB-producing strain were first grown in LB medium. Cells were then adapted to the deuterated Enfors minimal medium with the following composition: 6.86 g l⁻¹ $(NH_4)_2SO_4$, 1.56 g l⁻¹ $KH_2PO_4$, 6.48 g l⁻¹ $Na_2HPO_4 \cdot 2H_2O$, 0.49 g l⁻¹ $(NH_4)HC_6H_5O_7$ (diammonium hydrogen citrate), 0.25 g l⁻¹ $MgSO_4 \cdot 7H_2O$, with 1.0 ml l⁻¹ of trace metal stock solution (0.5 g l⁻¹ $CaCl_2 \cdot 2H_2O$, 16.7 g l⁻¹ $FeCl_3 \cdot 6H_2O$, 0.18 g l⁻¹ $ZnSO_4 \cdot 7H_2O$, 0.16 g l⁻¹ $CuSO_4 \cdot 5H_2O$, 0.15 g l⁻¹ $MnSO_4 \cdot 4H_2O$, 0.18 g l⁻¹ $CoCl_2 \cdot 6H_2O$, 20.1 g l⁻¹ EDTA), 5 g l⁻¹ glycerol-$d_8$ (Eurisotop). A single colony of *E. coli* containing the pET29b(+)-pa2l plasmid grown on an LB agar plate supplemented with 50 µg ml⁻¹ kanamycin was used to inoculate 15 ml of LB medium with kanamycin. The preculture was then used to inoculate 15 ml of the hydrogenated Enfors minimal medium to $OD_{600}$ of 0.1 and grown overnight. This preculture was used to inoculate 15 ml of the deuterated Enfors minimal medium to $OD_{600}$ of 0.1 and grown overnight. This step was repeated 5 times prior to the fed-batch fermentation.

**Deuterated fed-batch fermentation of LecB**. A final preculture of 150 ml was used to inoculate 1.2 l of deuterated Enfors minimal medium in a 3 l bioreactor (Infors, Switzerland). The pD of the culture medium was regulated at 7.2 by addition of 4% NaOD (Eurisotop, France). The temperature was maintained at 30 °C. After consumption of the deuterated glycerol-$d_8$ from the culture medium, the fed-batch phase was initiated by continuous feeding with the additional 30 g of the deuterated glycerol-$d_8$. The protein expression was induced overnight (17 h) with 1 mM isopropyl-β-D-thiogalactopyranoside (IPTG) at the $OD_{600}$ of 13 at 30 °C. After the fermentation, the cells were recovered by centrifugation (10,500 × g for 1 h at 6 °C) and the wet cell paste was frozen at −80 °C for long-term storage. The final yield was 51 g of the deuterated cell paste from a final volume of 1.6 l culture medium.

**Protein purification**. Deuterated LecB was purified by affinity chromatography using the NGC system (Bio-Rad, Marnes-la-Coquette, France) on a 10 ml mannose-Sepharose resin packed in the C10/10 column. All buffers used for purification were prepared in $H_2O$. Cells were resuspended in buffer A (20 mM Tris-HCl, pH 7.5, 100 mM NaCl, 100 µM $CaCl_2$) in the presence of an EDTA-free protease inhibitor cocktail (cOmplete™, Roche) and treated with DENARASE® endonuclease (c-LEcta GMBH, Leipzig, Germany). Cells were lyzed using a cell disruptor with pressure of 1.8 kbar (Constant Systems Ltd., Northants UK). After centrifugation (24,000 × g for 30 min at 4 °C), the supernatant was filtered (0.45 µm) and the clear cell lysate was loaded onto 10 ml mannose-agarose column pre-equilibrated with the buffer A. After extensive washing the unbound proteins, the deuterated LecB was eluted with buffer A containing 100 mM free D-mannose. The purity of the protein was examined by 16% Tris-tricine SDS-PAGE stained with Coomassie Blue. The fractions containing the pure protein were pooled together and dialyzed against 5 l of buffer A at 4 °C for a week with changing the buffer once per day. The protein was concentrated using a 5 kDa cut-off concentrator (Corning® Spin-X® UF, England) and flash-frozen in liquid nitrogen for long-term storage. The typical yield of the deuterated LecB was about 4 mg of protein per gram of wet cell paste.

**Isothermal titration calorimetry**. All measurements were carried out at 25 °C using a MicroCal iTC200 isothermal titration calorimeter (Microcal-Malvern Panalytical, Orsay, France). Samples were dissolved in 20 mM Tris buffer, pH 7.5 with 100 mM NaCl and 100 µM $CaCl_2$ prepared in $H_2O$ and $D_2O$, respectively. A total of 20 injections of 2 µl of fucose (4 mM L-fuc/7.6 mM L-Fuc-$d_{12}$) solution each were injected at intervals of 120 s with constant stirring of 750 rev min⁻¹ in the 200 µl sample cell containing the LecB protein (0.46 mM LecB/0.50 mM D-LecB). Experimental data were fitted to the theoretical titration curve with the one set of sites model in the software supplied by MicroCal with ΔH (enthalpy change), Ka (association constant), and $n$ (number of binding sites per monomer) as adjustable variables. The free energy change (ΔG) and entropy contributions (TΔS) were calculated from the equation $\Delta G = \Delta H - T\Delta S = -RT \ln K_a$ ($T$ is the absolute temperature and $R = 8.314$ J K⁻¹ mol⁻¹). Two independent titrations were performed for each experiment.

**Crystallization**. The perdeuterated LecB lectin was crystallized using a vapor-diffusion sitting drop method. The protein solution of 10 mg ml⁻¹ was incubated with 1.4 mM of perdeuterated fucose in the presence of 100 µM $CaCl_2$ during 1 h at RT prior to crystallization. Perdeuterated fucose-$d_{12}$ was produced by glyco-engineered *E. coli* in a high cell-density culture as reported previously[29]. Co-crystals of deuterated LecB with L-fucose-$d_{12}$ were obtained in the following conditions: 0.1 M Tris/DCl, pD 7.1, 20% (w/v) PEG 4000. The protein was mixed with the reservoir solution in 1:1 ratio and incubated at 19 °C.

**Neutron data collection and processing**. Neutron quasi-Laue diffraction data from the crystal of perdeuterated LecB/fucose complex were collected at RT using the LADI-III diffractometer[40] at the Institut Laue-Langevin in Grenoble using a crystal of perdeuterated LecB in complex with perdeuterated fucose-$d_{12}$ with volume of ~0.1 mm³. A neutron wavelength range ($\Delta\lambda/\lambda = 30\%$) of 2.8–3.8 Å was used for data collection with diffraction data extending to 1.9 Å resolution. The crystal was held stationary at different φ (vertical rotation axis) for each exposure. A total of 24 images were recorded (18 h per exposure) from two different crystal orientations. The neutron diffraction images were processed using the *LAUEGEN* software[41]. The *LSCALE* program[42] was used to determine the wavelength normalization curve using intensities of symmetry-equivalent reflections measured at different wavelengths. The data were merged and scaled using *SCALA*[43].

**X-ray data collection and processing**. RT X-ray data were collected from the same crystal as used for the neutron data collection. The X-ray data were recorded on the GeniX 3D Cu High Flux diffractometer (Xenocs) at the Institut de Biologie Structurale in Grenoble, France. The data were processed using the *iMosflm* software[44], scaled and merged with *AIMLESS*[45] and converted to structure factors using *TRUNCATE* in the *CCP4* suite[46].

Smaller crystals of deuterated LecB co-crystallized with deuterated fucose were harvested manually and cryo-cooled at 77 K by flash-cooling in liquid nitrogen. X-ray datasets were collected at the SOLEIL synchrotron (Saint Aubin, France) on the PROXIMA-1 beamline. Images were recorded on the EIGER-X 16 M detector (Dectris Ltd., Baden, Switzerland) and processed by *XDS*[47].

**Joint neutron and X-ray refinement**. The initial model (PDB: 1GZT) with water molecules, metal ions and ligands removed was used for the molecular replacement using *PHASER* in *PHENIX* suite. All further data analysis was done using the *PHENIX* package[48]. Refinement was performed using a restrained maximum-likelihood method in *phenix.refine*[49]. Restraint files for deuterated fucose were generated by *eLBOW* utility[50] in *PHENIX*. Ligands were placed manually in *COOT*[51]. Water molecules were introduced automatically in *phenix.refine* based on the positive peaks in the $mF_o$-$DF_c$ neutron scattering length density and inspected manually. Deuterium atoms were introduced using the *Readyset* utility in *Phenix* and refined individually. All water molecules were modeled as $D_2O$. Water oxygen atoms were initially placed according to the electron density peaks. The orientations of water molecules were refined and modified based on the potential hydrogen donor and acceptor orientation.

The neutron $R_{work}$ and $R_{free}$ values for the final model were 19.1% and 24.6%, respectively, while the X-ray $R_{work}$ and $R_{free}$ values were 10.4% and 14.2%, respectively. *MolProbity*[52] software was used for structure validation. Refinement statistics are presented in Table 1 and the completeness curves for the X-ray structure in Supplementary Fig. 6. The models with the diffraction data have been deposited in the Protein Data Bank under accession codes 7PRG for the X-ray/neutron jointly refined model and 7PSY for the 0.9 Å 100 K X-ray structure. Molecular figures were prepared using PyMOL (Schrödinger).

**Reporting summary**. Further information on research design is available in the Nature Research Reporting Summary linked to this article.

## Data availability

Coordinates and structure factors generated in this study have been deposited in the Protein Data Bank under accession codes: 7PRG for the room temperature X-ray/neutron structure of the perdeuterated LecB/fucose complex and 7PSY for the 100 K X-ray structure of perdeuterated complex. All other structures of LecB lectin used for comparisons were taken from the Protein Data Bank under accession codes: 1GZT, 1OXC, and 1UZV.

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

## Acknowledgements

The authors wish to acknowledge the ILL for provision of beamtime on LADI-III and technical support. We thank the ILL for the provision of studentship funding to L.G.

V.T.F. acknowledges the UK Engineering and Physical Sciences Research Council EPSRC) for grants GR/R99393/01 and EP/C015452/1 that funded the creation of the Deuteration Laboratory within ILL's Life Sciences Group. A.I. acknowledges support from Glyco@Alps (ANR-15-IDEX02) and Labex Arcane/CBH-EUR-GS (ANR-17-EURE-0003). This work was supported by access to the HTX lab facility at EMBL and the PSB. We wish to acknowledge the IBS for access to the X-ray diffractometer and Proxima-1 beamline at SOLEIL Synchrotron, Saint Aubin, France for provision of beamtime. We wish to thank Annabelle Varrot and Sakonwan Kuhaudomlarp for their help in the X-ray data collection and Emilie Gillon for her help in the ITC measurements.

## Author contributions

A.I., M.H., J.M.D., M.P.B., and V.T.F. designed the experiment. L.G., J.M.D., M.H., and V.T.F. produced the deuterated biomolecules. L.G. purified the various components and carried out the experimental work. M.P.B. collected and processed the neutron data and provided expertise in structure determination. L.G. solved the crystal structures and refined them. L.G. prepared the figures and wrote the manuscript with A.I. and J.M.D., and with critical inputs from all authors.

## Competing interests

The authors declare no competing interests.
