## [Peer Review File · Nature Communications]

REVIEWER COMMENTS

Reviewer #1 (Remarks to the Author):

This paper provides information about the mechanism of binding of sugar ligands to LecB, a lectin from *Pseudomonas aeruginosa*. The work is of interest because LecB is involved in the bacterium binding to host cells. Although previous high resolution structures determined by X-ray crystallography have already shown that the bound monosaccharide is ligated to 2 calcium ions, here the structure of a deuterated complex of LecB bound to fucose, determined by neutron crystallography, provides interesting new details of the binding site that may account for its unusually high affinity for monosaccharides compared to sugar binding sites in other lectins.

The new structure, which shows the location of all hydrogen atoms, indicates that one of the H-bonds between OH groups of fucose and the protein is a low barrier hydrogen bond. It also shows that none of the 6 carboxyl groups involved in ligating the two calcium ions is protonated, indicating that the binding site is negatively charged. Comparison of the neutron structure, determined at room temperature, with a previous X-ray structure determined at 100K shows details of thermal motion of water molecules in the binding site, which accounts for the favourable entropy contribution to binding. This is the first determination of a neutron structure for a lectin from a human pathogen and it provides significant new insight into the mechanism of ligand binding.

The work is thorough, with an X-ray structure of the deuterated LecB-fucose complex determined as well as the neutron structure, to allow careful comparisons. In addition, isothermal titration calorimetry was used to confirm that deuteration of LecB and fucose did not change the thermodynamics of binding when compared to binding of fucose to hydrogenated LecB.

The paper is generally clearly written, but a few revisions would help with presentation of data:

- 1.) Although each calcium is hepta-coordinated, Fig 3 shows only 6 coordination bonds to each. If all 7 can't be shown, the legend should explain which one is not shown.
- 2) Fig 4 is not very clear. It is very hard to see the low barrier hydrogen bond in (a) - perhaps these figures could be made larger. It is also not very easy to see which atoms are involved in the six-membered rings in (b), particularly on the left hand image.
- 3) The Introduction is rather vague about the role of LecB in *P. aeruginosa* pathogenicity, just indicating that it is involved. A few sentences, giving details of the exact contribution of LecB to adhesion and/or biofilm formation would be helpful.

Reviewer #2 (Remarks to the Author):

The manuscript by Gájdos et al. describes crystal structures of the lectin LecB from *Pseudomonas aeruginosa* in complex with fucose, determined using neutron and very high-resolution X-ray crystallography. LecB is an important lectin for the infectivity of *P. aeruginosa* and inhibitor design against LecB is thus of potential pharmaceutical interest. For carbohydrate-based ligands and inhibitors, a detailed understanding of the positions of hydrogen atoms and the directionality of hydrogen bonds is very important for future drug design.

The technical quality of the work is very high. The diffraction data are excellent (especially given the small size of the crystal used for neutron data collection) and the structures are fastidiously determined and refined. The use of perdeuterated fucose is an important technical advance allowing full resolution of all hydrogen atoms on the ligand, avoiding the cancellation effects normally caused by the presence of hydrogen at the aliphatic positions. The affinity of the hydrogenated and perdeuterated proteins was determined using ITC to make sure that there were no effects on ligand binding energetics, and no significant differences were seen. The work is well-described and would allow the experiments to be reproduced.

One important conclusion is that the proximity of the two Ca²⁺ ions is important for the orientation of the hydroxyl group on C3 of fucose and that this could be an important contributor to the relatively high affinity of LecB for fucose (low micromolar rather than the usual mM affinity of lectins for individual sugar units).

Comments:

I feel that there is a disconnect between the first and second paragraphs of the Introduction on p.3. A statement of the specific purpose of the present study would fill the gap here.

PDB validation reports: As noted, both structures are of excellent quality, but why is the MolProbity clashscore (no. of bad contacts per 10 000 atoms) worse for the 0.9 Å cryo structure than for the 1.85 Å joint neutron/X-ray structure (6 vs. 1)? One would expect that if the neutron structure was refined against the high-resolution X-ray data, the low number of clashes would be retained. Or does it have to do with a higher number of water molecules in the 100K structure?

I would argue that it's misleading to give the resolution of the 100K structure as 0.9 Å since the completeness of the data between 0.92-0.90 Å is only 25%. From such simple numbers it's very hard to work out how quickly the completeness falls off with resolution, so the authors should provide more detail, at least at what resolution the data are still ~100% complete but perhaps a completeness curve.

How many more water molecules were identified in the 100K structure in comparison to the RT structure?

Minor points:

On p. 3: "Carbohydrate chemists have proposed several fucose or mannose derivatives" – as what? I guess inhibitors.

On p. 6: I think the electron density shape is better described as "pyramidal". Also, the authors should confirm that the general statements made in this paragraph are true for all four monomers or specify where they are not.

On p. 7: The side chains of D99 and the C-terminal carboxylate group of G199 are shown with "valence representation" in PyMOL which means that they have been drawn with determinate double and single bonds. As a result, the sp³ oxygen atom that COULD be protonated (but which is not) is the one sharing the hydrogen atom with the respective fucose hydroxyl group rather than the sp² carbonyl oxygen. It would be better to flip these 180° or not to use the valence representation.

On p. 9 the authors state that "The protonation states of these six acidic groups is a crucial question" without having justified why it's crucial. Could they elaborate?

Also on p. 9: The authors say that the Ca²⁺ ions are coordinated by among others "...the carboxylate atoms of six acidic amino acids (three aspartates, one glutamate and the C-terminal group of Gly from the neighbouring monomer). I make that only four acidic residues.

In Figure 4, I find the size of the Ca²⁺ ions in panel a too large (though perhaps realistic!) Also, I have difficulty seeing the 6-membered rings that are mentioned. Counting only the oxygen atoms and Ca²⁺ I get 5 members, if counting all the connected atoms I get 8. Can the authors clarify? Finally, I find the use of the word "synergistic" too specific. This statement implies that they truly have a cooperative effect on the energetics of binding, but as far as I can see this hasn't been experimentally demonstrated, it is just a hypothesis for now. The word is used again twice in the section "Occurrence of synergistic rings of contacts".

On p.12, why do the authors think that the standard deviations for the bond lengths in the low barrier H-bonds are so much higher than for the classical H-bonds?

On p. 15, I don't understand why the word "therefore" is used in the second sentence, as I don't see that the two sentences follow on from each other logically.

p. 19, Crystallization: the concentration of perdeuterated fucose would be better expressed in M.

On p. 20, I would say restrained maximum-likelihood TARGET FUNCTION". Later on, the words "quasi-

Laue" can be removed, as they pertain to the diffraction data and not to the model. I don't think that the method used to collect neutron data has a quantifiable effect on the model.

We thank the reviewers for their comments and their helpful suggestions. All of the comments were considered. The detailed answers are listed below and the revised manuscript include all modifications marked in red.

Referee: 1

We thank the reviewer for the positive comments of our manuscript and constructive suggestions for improving the presentation of the data.

1. Although each calcium is hepta-coordinated, Fig 3 shows only 6 coordination bonds to each. If all 7 can't be shown, the legend should explain which one is not shown.

Answer. The figure has been improved as suggested to show all seven coordination bonds.

2. Fig 4 is not very clear. It is very hard to see the low barrier hydrogen bond in (a) - perhaps these figures could be made larger. It is also not very easy to see which atoms are involved in the six-membered rings in (b), particularly on the left hand image?

Answer. The figure has been modified to increase the visibility of the low-barrier hydrogen bond as well as the six-membered rings.

3. The Introduction is rather vague about the role of LecB in *P. aeruginosa* pathogenicity, just indicating that it is involved. A few sentences, giving details of the exact contribution of LecB to adhesion and/or biofilm formation would be helpful.

Answer. As suggested, two sentences have been added in the introduction, for better description of the involvement of LecB in different steps of the infection process.

Referee: 2

We thank the reviewer for a thorough evaluation of our manuscript and for suggestions that helped us improving the final version.

1. I feel that there is a disconnect between the first and second paragraphs of the Introduction on p.3. A statement of the specific purpose of the present study would fill the gap here.

Answer. The connection has been improved by adding a new sentence beginning of the second paragraph.

2. PDB validation reports: As noted, both structures are of excellent quality, but why is

the MolProbity clashscore (no. of bad contacts per 10 000 atoms) worse for the 0.9 Å cryo structure than for the 1.85 Å joint neutron/X-ray structure (6 vs. 1)? One would expect that if the neutron structure was refined against the high-resolution X-ray data, the low number of clashes would be retained. Or does it have to do with a higher number of water molecules in the 100K structure?

Answer. Indeed the RT joint neutron/X-ray structure has a lower clashscore. In this case, the structure was refined against X-ray data extending to 1.85 Å resolution and neutron data extending to 1.9 Å resolution, both collected at RT on the same crystal. The structure contains fewer alternative conformations of amino acids and fewer water molecules (364). The 100K X-ray structure was refined against data extending to 0.9 Å resolution. This structure contains more alternative conformations of amino acid residues as well as a higher number of water molecules (774 vs 364 in the jointly refined structure) which caused the higher value of the clashscore.

3. I would argue that it's misleading to give the resolution of the 100K structure as 0.9 Å since the completeness of the data between 0.92-0.90 Å is only 25%. From such simple numbers it's very hard to work out how quickly the completeness falls off with resolution, so the authors should provide more detail, at least at what resolution the data are still ~100% complete but perhaps a completeness curve

Answer. We agree with the reviewer that the completeness is very low in the highest resolution shell. When examining different cut-off values of 0.95 Å and 1.0 Å resolution, it could be seen that the completeness was improving rapidly from 76 % to 95 % respectively. Nevertheless, we decided to keep the data to 0.9 Å resolution since the CC1/2 value of 66 % suggested the data were of good quality. Here we provide the completeness vs. resolution curve for the 0.9 Å data.

%poss is completeness in the shell

C%poss in cumulative to that resolution

The anomalous completeness values (**AnomCmpl**) are the percentage of possible anomalous differences measured

AnomFrc is the % of measured acentric reflections for which an anomalous difference has been measured

4. How many more water molecules were identified in the 100K structure in comparison to the RT structure?

Answer. As mentioned above, the RT structure contains 364 water molecules while the 100K structure contains 774 water molecules. This information has been added in Table 1.

Minor points:

5. On p. 3: “Carbohydrate chemists have proposed several fucose or mannose derivatives” – as what? I guess inhibitors.

Answer. Clarified by adding “as high-affinity ligands”.

6. On p. 6: I think the electron density shape is better described as “pyramidal”. Also, the authors should confirm that the general statements made in this paragraph are true for all four monomers or specify where they are not.

Answer. Text has been corrected as suggested.

7. On p. 7: The side chains of D99 and the C-terminal carboxylate group of G199 are shown with “valence representation” in PyMOL which means that they have been drawn with determinate double and single bonds. As a result, the sp³ oxygen atom that COULD be protonated (but which is not) is the one sharing the hydrogen atom with the respective fucose hydroxyl group rather than the sp² carbonyl oxygen. It would be better to flip these 180° or not to use the valence representation.

Answer. Thank you for pointing this out. Figure 2 has been modified accordingly.

8. On p. 9 the authors state that “The protonation states of these six acidic groups is a crucial question” without having justified why it’s crucial. Could they elaborate?

Answer. This is important indeed for clarifying the mechanisms for high affinity. It has been clarified.

9. Also on p. 9: The authors say that the Ca²⁺ ions are coordinated by among others “...the carboxylate atoms of six acidic amino acids (three aspartates, one glutamate and the C-terminal group of Gly from the neighbouring monomer). I make that only four acidic residues.

Answer. Thanks for pointing out this mistake. There are six carboxylate groups of amino acids in the binding site (D96, D99, D101, D104, E95 and the C-terminal group of G114), five coordinating calcium and one bound to fucose (D96). This has been clarified in the text page 9 with more details on the amino acids involved.

10. In Figure 4, I find the size of the Ca²⁺ ions in panel a too large (though perhaps realistic!) Also, I have difficulty seeing the 6-membered rings that are mentioned. Counting only the oxygen atoms and Ca²⁺ I get 5 members, if counting all the connected atoms I get 8. Can the authors clarify?

Answer. The figure has been modified to ease the visibility of the low-barrier hydrogen bonds as well as the six-membered rings.

11. Finally, I find the use of the word “synergistic” too specific. This statement implies that they truly have a cooperative effect on the energetics of binding, but as far as I can see this hasn’t been experimentally demonstrated, it is just a hypothesis for now. The word is used again twice in the section “Occurrence of synergistic rings of contacts”.

Answer. Indeed, we observed rings of contact but the “synergistic” effect is only a hypothesis, that fits well with previous calculations. The term has been corrected to “proposed synergistic” or removed through the text.

12. On p.12, why do the authors think that the standard deviations for the bond lengths in the low barrier H-bonds are so much higher than for the classical H-bonds?

Answer. The hydrogen in the low barrier hydrogen bond is not “fixed” at mid-distance between donor and acceptor and the peak is at different distance depending of the monomer (Table S2) but always at longer distance than classical h-bond. One can expect a higher standard deviation since the character of the bond includes a higher mobility of the hydrogen atom.

13. On p. 15, I don’t understand why the word “therefore” is used in the second sentence, as I don’t see that the two sentences follow on from each other logically.

Answer. We agree that the second “therefore” should be removed.

14. p. 19, Crystallization: the concentration of perdeuterated fucose would be better expressed in M.

Answer. Corrected as suggested.

15. On p. 20, I would say restrained maximum-likelihood TARGET FUNCTION”. Later on, the words “quasi-Laue” can be removed, as they pertain to the diffraction data and not to the model. I don’t think that the method used to collect neutron data has a quantifiable effect on the model.

Answer. The sentences have been modified.

REVIEWERS' COMMENTS

Reviewer #2 (Remarks to the Author):

The authors have addressed all my questions and comments on the original manuscript and it is improved overall. I have only three comments on the revised version and the authors' explanations.

I accept the argument that the 0.9 Å structure contained more alternative conformations and water molecules than the one at 1.85 Å, but by the same argument, some of these must have been poorly modelled, or they would not have resulted in a higher clashscore. Higher resolution structures are supposed to have better geometry! But this is just a comment, not a request for revision.

Regarding the completeness graph for the data, thanks to the authors for providing this. However, I meant that it should/could be included in the article as supplementary information.

In the added sentence on pathogenicity in the introduction, "in mouse infection model" should be "in a mouse infection model" or "in mouse infection models", depending on how many models there are.

"pyramidal-shaped" should be either "pyramidal" or "pyramid-shaped".

We thank reviewer 2 for his/her careful reading and we corrected the manuscript according to the suggestions.

1. The authors have addressed all my questions and comments on the original manuscript and it is improved overall. I have only three comments on the revised version and the authors' explanations.

I accept the argument that the 0.9 Å structure contained more alternative conformations and water molecules than the one at 1.85 Å, but by the same argument, some of these must have been poorly modelled, or they would not have resulted in a higher clashscore. Higher resolution structures are supposed to have better geometry! But this is just a comment, not a request for revision.

Answer. We fully agree with the comment – no revision needed.

2. Regarding the completeness graph for the data, thanks to the authors for providing this. However. I meant that it should/could be included in the article as supplementary information.

Answer. As suggested, the completeness graph is now included as Figure S6 in supplementary information and cited in the last paragraph of the Method section of the manuscript.

3. In the added sentence on pathogenicity in the introduction, “in mouse infection model” should be “in a mouse infection model” or “in mouse infection models”, depending on how many models there are

Answer. Grammar has been corrected.

4. “pyramidal-shaped” should be either “pyramidal” or “pyramid-shaped”.

Answer. Thanks for correction, “pyramid-shaped” sounds indeed better.